# Control and Diagnostics System Generator for Complex FPGA-Based Measurement Systems

**DOI:** 10.3390/s21217378

**Published:** 2021-11-06

**Authors:** Wojciech M. Zabołotny, Marek Gumiński, Michał Kruszewski, Walter F.J. Müller

**Affiliations:** 1Faculty of Electronics and Information Technology, Institute of Electronic Systems, Warsaw University of Technology, Nowowiejska 15/19, 00-65 Warszawa, Poland; m.guminski@elka.pw.edu.pl (M.G.); m.kruszewski@elka.pw.edu.pl (M.K.); 2GSI-Helmholtzzentrum für Schwerionenforschung GmbH, Planckstraße 1, 64291 Darmstadt, Germany; w.f.j.mueller@gsi.de

**Keywords:** FPGA, Wishbone, control interface, VHDL, system management, system diagnostics

## Abstract

FPGA-based data acquisition and processing systems play an important role in modern high-speed, multichannel measurement systems, especially in High-Energy and Plasma Physics. Such FPGA-based systems require an extended control and diagnostics part corresponding to the complexity of the controlled system. Managing the complex structure of registers while keeping the tight coupling between hardware and software is a tedious and potentially error-prone process. Various existing solutions aimed at helping that task do not perfectly match all specific requirements of that application area. The paper presents a new solution based on the XML system description, facilitating the automated generation of the control system’s HDL code and software components and enabling easy integration with the control software. The emphasis is put on reusability, ease of maintenance in the case of system modification, easy detection of mistakes, and the possibility of use in modern FPGAs. The presented system has been successfully used in data acquisition and preprocessing projects in high-energy physics experiments. It enables easy creation and modification of the control system definition and convenient access to the control and diagnostic blocks. The presented system is an open-source solution and may be adopted by the user for particular needs.

## 1. Introduction

Modern measurement systems, especially those used in high-energy physics or plasma physics experiments, require complex data acquisition and concentration systems to efficiently collect data from multiple input channels. They often require high-speed data processing to reduce the volume of the received data stream, such as selection of interesting events, aggregation, or compression [1,2,3]. Those systems typically use sensors connected via frontend electronics boards (FEBs) using specialized high-speed interfaces [4,5,6]. Additionally, perfect synchronization of data streams received in different channels is required [7]. The programmable devices—Field Programmable Gate Arrays (FPGA) are usually used to provide those functionalities [8,9,10,11,12,13,14]. Their big advantage is high flexibility, enabling significant changes to the communication protocols or data processing algorithms without modifying the underlying hardware.

### 1.1. General Properties of FPGA-Implemented Data Acquisition and Preprocessing Systems

Such complex systems must be implemented in a modular way where standard and well-tested basic blocks are used to implement more advanced functionalities. This pattern is repeated over the functional levels, resulting in a multilayer, hierarchical block structure.

Reuse of standard blocks in various systems and for various purposes imposes their parameterization. The complexity of the design may change during development and debugging. For example, a simpler version with a limited number of input channels may be used at the beginning of the development to help to detect, isolate and fix bugs. Therefore, the number and structure of higher-level blocks should also be parameterized.

The data processed by such a measurement system may have a complex structure. The HDL language used for implementation should be capable of handling such data efficiently. They can be described with VHDL records [15] or with SystemVerilog structures. However, VHDL seems to be better suited for the implementation of such complex and parameterized systems. Its strict type checking provides good mistakes and errors detection when the system is modified.

To summarize, the data acquisition and preprocessing system is a complex, parameterized, multilayer hierarchical design, significantly relying on advanced features of the VHDL language. The main functionalities, data collection and preprocessing, are usually implemented as a pipelined datapath, optimized for maximal throughput and minimal latency.

Except for the datapath, such a system also requires an efficient control and diagnostics layer. Its purpose is to configure the data processing part, control it, and receive information about its operation (for example, the warnings and errors, the performance metrics, etc.).

The Control and Diagnostics (C&D) system is usually decoupled from the datapath (Certain information about the system’s operation and processing parameters may also be added to the output data stream as metadata. That may help in further data analysis.) and consists of registers accessible via a separate control bus.

The read and write operations on those registers are performed by the software running on a CPU (The CPU may be implemented inside the FPGA (so-called soft CPU, such as Xilinx Microblaze or Forth CPU—J1B) or an external device communicating with FPGA via an appropriate interface.). Therefore, an essential feature of the C&D-system is to provide convenient access from the software to selected control and diagnostics registers. The control and diagnostic operations usually use random access to the registers using read, write, and read-modify-write operations. Long block transfers are used relatively rarely.

The typical structure of such an FPGA-implemented measurement system is shown in Figure 1.

### 1.2. Requirements for the Control and Diagnostics (C&D) System

The described properties of the data acquisition and preprocessing systems are the basis for formulating requirements for the C&D-system. Those requirements are used to evaluate existing solutions (see Section 1.4 and Table 1).

#### 1.2.1. Selection of the Control Bus

Currently, the most popular internal buses for FPGA designs are the Wishbone [16] and AXI4 [17]. The selection of the control bus depends on multiple criteria and must consider the specific requirements of the control and diagnostic system.

Wishbone is a relatively old standard–version B3 [18] was released in 2002, and version B4 [19] in 2010. It is open and is very popular in the open-source world. There are many open implementations of Wishbone masters and slaves [20,21,22]. It offers various modes of operation, where the “classic standard single read/write mode” enables very simple implementation of the slaves. The Wishbone bus is also compatible with the IPbus [23] solution, used to control FPGA-based boards via Ethernet.

The AXI4 bus offers excellent block transfer performance but requires a more complex implementation of the bus slaves [24]. It also offers a simplified “AXI-Lite” [25] version suited explicitly for accessing the memory-mapped registers, but it is still more complex than Wishbone [26].

Because the control and diagnostics system mainly uses random accesses to the registers, and the resource use by the C&D-system should be minimized, the Wishbone bus seems to be the right solution. That decision is also supported by the broad availability of Wishbone-compatible slave blocks. The 32-bit width of data and address buses should be sufficient. The synthesis tools should optimize unused address lines if the C&D-system uses a smaller subset of the address space.

#### 1.2.2. Requirements for the Registers

Exchange of the control and diagnostic information is achieved by reading and writing registers. Two main types of registers may be defined: the control registers (available for reading and writing) and the status registers (that can be only read) (In fact, it could be possible to implement also “write-only” registers. They should be backed by the “shadow registers” implemented in the software and holding the last written value. The usage of “write-only” registers may further limit resource consumption. However, they do not allow verification of the written value—a helpful feature in the debugging mode. Therefore, we decided not to implement them.).

If not specified otherwise, the registers have a width (number of bits) equal to the width of the data bus in the control system. However, if the control parameter or the status value to be transferred via the register has a smaller width, it is desirable that the width of the register could be limited accordingly.

On the other hand, sometimes, it is useful to combine multiple low-width control parameters or status values into a single register. That gives a possibility to read or write them simultaneously and reduces resource consumption. For that purpose, the C&D-system should support “bit-fields” described by their width and position of their least significant bit in the register.

#### 1.2.3. Support for the Hierarchical Parameterized Design

As described in Section 1.1, the data acquisition and processing system usually has a multilayer hierarchical structure. That affects the requirements for the control system. The hierarchy may be extended horizontally—there may be multiple identical functional blocks or multiple identical control or status registers. Therefore, the C&D-system should support vectors of blocks and vectors of registers.

The multilayer hierarchy assumes that the blocks may be nested, and the C&D-system should also reflect it. In that case, the connections to the nested blocks must be easy to create and maintain.

The parameterization functionality described in Section 1.1 requires that the lengths of the mentioned vectors and the presence of the particular blocks and registers should be defined by modifiable parameters.

### 1.3. Need for C&D System Generator

The structure of the control and diagnostic system is tightly coupled with the structure of the measurement system. That means that usually, it must be created for the particular system and evolve together with it.

All registers must have addresses assigned in the address space of the control bus. To simplify the addressing, the vectors of registers should occupy the contiguous areas in the address space.

Similarly, each block must have assigned its private area in the bus address space, and the blocks belonging to a vector should occupy the consecutive areas.

The proper alignment of blocks simplifies the address decoding and enables reusing resources between address decoders (In the optimal solution, the base address of each block should be aligned to the $2^{N}$ boundary, where $2^{N}$ is the smallest power of 2 that may fit all the addresses belonging to the block. Such alignment allows connecting the block’s internal address decoder only to address bits 0 to N−1.).

The above means that the **address map** of the C&D-system may require modification after each change of the system’s structure or even after changing the parameters describing that structure. Therefore, an automated system for address assignment is necessary.

The assigned addresses must also be somehow passed to the C&D software. So, we need a **C&D-system generator** capable of assigning the addresses and generating the address map in a format legible for hardware synthesis tools and software environments.

It is also desirable that the generator implements the registers and the control bus infrastructure, minimizing the effort needed to integrate the generated C&D-system with the rest of the data acquisition and processing system.

### 1.4. Possible Existing Solutions for C&D System Generation

Generation of C&D-systems for FPGA-implemented systems is not a new problem. It has been investigated for many years, and many such solutions have been developed.

#### 1.4.1. SystemRDL

The most advanced system related to the generation of the C&D-systems is SystemRDL [27]. The SystemRDL is a language aimed at the detailed description of the registers. It tries to cover all possible aspects of register structure and behavior, including descriptions of an arbitrarily complex hierarchy of blocks and registers. The last version supports the parameterization of components and the structure of the system. SystemRDL is well designed and mature but also very complex. When generating a C&D-system, it is necessary to use a special tool to translate the SystemRDL description into the output format - the HDL implementation or software source supporting the communication. Unfortunately, currently, there are only a few such tools available. Agnisys offers a commercial solution [28], but it is closed-source and cannot be modified by the user. There is an open-source **ordt** from Juniper [29], but it does not generate VHDL code. It does not support the Wishbone bus either. There is a whole set of SystemRDL related tools developed in the GitLab repository [30]. Unfortunately, up to now, there is no tool capable of generating the VHDL output.

#### 1.4.2. Internal Interface and Component Internal Interface

The Internal Interface (II) was developed by Krzysztof Pozniak and others for electronic systems created for CMS and DESY [31,32,33]. It has been later extended with object functionalities, forming the Component Internal Interface (CII) [34,35]. II initially was using a VME-like interface as a local FPGA bus. In the CII version, it has been supplemented with a possibility to control the Wishbone bus. The CII-implemented C&D-systems may work with software written in C++, Java, and Matlab. II/CII supports complex data structures, such as arrays of arbitrary lengths, vectors of bits of arbitrary lengths, etc. However, that flexibility has its price: a high complexity of the interface, significant resource consumption, and lower maximum clock frequency. Another disadvantage of the CII is the high complexity of the description of the registers and the fact that it is a closed solution that cannot be used in open-source projects.

#### 1.4.3. Address Generator for IPbus

During the development of the Data Processing Boards (DPB) for the CBM experiment [36], our team faced the situation where manual allocation of register addresses for IPbus-connected C&D-system became inefficient due to increase of complexity of the developed firmware.

The **addr_gen** [37,38] system was proposed to cure that problem. It accepts the description of the blocks and registers in Python language. It also supports the hierarchy of blocks and vectors of registers and blocks.

The system’s output is two VHDL packages defining the constants with parameters and a complex VHDL record with addresses of particular blocks, subblocks, and registers. Integration with software is supported by generating a Python dictionary with the list of assigned addresses and, for C++, the IPbus XML Address Map [39].

That tool does not support bit-fields and does not generate the HDL code for accessing the registers.

Handling all blocks and registers in a single component appeared to be extremely inconvenient. Routing of signals connected to those registers between the blocks and through the multiple hierarchy levels was messy and error-prone. To avoid it, registers should be located near the place where the connected signals are used. Therefore, the control bus infrastructure should be distributed between the blocks.

#### 1.4.4. Wishbone Slave Generator

One of the simplest C&D-system generators is the **wbgen2** application known as the **Wishbone slave generator** [40]. It supports the Wishbone local bus. The slave description is prepared in the C-like format and may contain registers, memory blocks, and FIFOs. The **wbgen2** generates the slave HDL code in VHDL or Verilog and C headers for integration with the software. Additionally, it may generate the documentation for the created slave in 
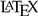
, Texinfo, or HTML. Unfortunately, it does not support vectors of registers or blocks, or nested blocks. The **wbgen2** is a free and open-source project. In fact, it was an inspiration for the development of the system described in this paper.

#### 1.4.5. Other Open-Source Tools

Other open-source tools include:**Opentitan register tool** [41], which unfortunately does not generate VHDL;**hdlregs** [42], which supports VHDL generation and even AXI4-Lite converter, but does not support a hierarchy of blocks;**RgGen** [43], which has limited support to hierarchical designs (only one level) and does not support Wishbone (only APB and AXI4-Lite buses);**rgen** [44], written in old Python 2.7 and supporting only IP-XACT input format; supports Wishbone, but only the old B3 version;**cheby** [45], which supports VHDL output, Wishbone bus, and hierarchy but does not support parameterization of the description.

### 1.5. Summary of the Review

The results of the review of existing solutions are summarized in Table 1. It can be seen that none of them fulfills all the requirements for the C&D-system generator formulated in Section 1.2. In the case of **cheby**, there was just a single functionality lacking. However, a quick review of the code has shown that adding this functionality may be difficult.

Therefore, the development of the new C&D-system generator was started (Please note that the development of AGWB was started in 2018, while the above review was done now. In fact, some of the listed tools were much less mature in 2018.).

## 2. Development of the C&D System Generator

The main task of the proposed C&D-system is the allocation of addresses for registers connected to the Wishbone bus, and the generation of appropriate HDL and software sources. Therefore, the system was named “**A**ddress **G**enerator for **W**ish**b**one”, in short **AGWB**. The version of the system from May 2019 is described in the conference proceeding paper [46]. Due to the limited size, the description was relatively scarce. Additionally, since that time, our system has been significantly improved based on experience from its use. This paper aims to provide a more detailed description of the concepts and implementation of the current version of **AGWB**.

### 2.1. The C&D System Hardware Structure

The general block structure of the C&D-system based on the requirements formulated in Section 1.2 is shown in Figure 2.

The core part is the Wishbone crossbar (The **xwb_crossbar** component from the General cores [22] library is used as the crossbar in each block.). The crossbar may be controlled by a single or multiple masters (see Section 2.2.1). The crossbar provides multiple child node buses—one for each connected slave. Each child node bus handles a certain exclusive segment of the overall address space. Allocation of those segments is handled by the address allocation algorithm, described in Section 2.3. The organization of the child node buses is shown in Figure 3. The implementation heavily relies on the VHDL records. Each Wishbone bus consists of two records—the first one for signals transmitted from master to slave and the second one for signals transmitted from slave to master. Another VHDL type is created to describe a vector of Wishbone buses.

A special case is the Wishbone bus controlling the registers. It is handled by an automatically generated process supporting reading and writing registers. The registers are always located at the beginning of the address segment occupied by the block (It is possible to reserve certain area below registers using the **reserved** attribute described in Section 2.2.2). The first two registers are always present and have a special meaning. They allow the software to verify if it communicates with the right C&D-system in the correct version. The first register is always the block ID (The value of block ID is calculated as a CRC32 value of its name.). The second register is always the block version (VER) (The value of VER is calculated as a CRC32 value of the combined XML configuration file used to generate the C&D-system. In designs using **variants**, the VER value for each block may be different (see Section 3.3)). Other registers are placed after the VER register (It is possible to place other special registers at addresses 4–7, using the **testdev_ena** attribute described in Section 2.2.2).

The described functionalities are implemented in the automatically generated “AGWB local node” (ALN) shown in Figure 4.

The ALN should be instantiated by the user in the design. Except for the VHDL code of ALN, AGWB also generates necessary VHDL packages supporting that instantiation. They are described more thoroughly in Section 3.

The structure of the generated C&D-system is defined by the user in the system definition, described in the next section.

### 2.2. AGWB Format for C&D System Definition

The C&D-system in AGWB is described with XML. The XML is often considered obsolete and bloated compared to newer alternatives such as JSON or YAML. However, it distinguishes between attributes and elements. That significantly facilitates the description of the C&D-systems. The XML format is defined with the RelaxNG schema, which allows automated detection of a significant part of errors (This is a functionality added in the new version of AGWB. The schema is also converted to RNC and DTD formats. Due to the bug in the library used to check compliance with RelaxNG and RNC schema, the DTD version is used for the checking.). The XML schema in the DTD format (selected as the most concise and legible) is shown in Listing 1. Most elements of the AGWB system definition have an obligatory **name** attribute defining their name and an optional **desc** attribute which may contain a description of the element and is passed to the generated HTML documentation.
sensors-21-07378-t001_Table 2Listing 1The AGBW XML schema presented in the DTD format. The number of possible elements and their attributes is relatively small.
<!ELEMENT sysdef (block|                          desc CDATA #IMPLIED
    constant|include)*>                           mode CDATA #IMPLIED
<!ATTLIST sysdef                                  reps CDATA #IMPLIED
  masters CDATA #IMPLIED                          stb CDATA #IMPLIED
  top CDATA #REQUIRED>                            stype CDATA #IMPLIED
                                                  type CDATA #IMPLIED
<!ELEMENT block (blackbox|creg|                   used CDATA #IMPLIED
    sreg|subblock)*>                              width CDATA #IMPLIED
<!ATTLIST block                                   ignore CDATA #IMPLIED>
  aggr_ins CDATA #IMPLIED
  aggr_outs CDATA #IMPLIED                      <!ELEMENT sreg (field)*>
  name CDATA #REQUIRED                          <!ATTLIST sreg
  reserved CDATA #IMPLIED                         ack CDATA #IMPLIED
  testdev_ena CDATA #IMPLIED                      desc CDATA #IMPLIED
  desc CDATA #IMPLIED                             mode CDATA #IMPLIED
  ignore CDATA #IMPLIED>                          name CDATA #REQUIRED
                                                  reps CDATA #IMPLIED
<!ELEMENT constant EMPTY>                         used CDATA #IMPLIED
<!ATTLIST constant                                stype CDATA #IMPLIED
  name CDATA #REQUIRED                            type CDATA #IMPLIED
  val CDATA #REQUIRED                             width CDATA #IMPLIED
  desc CDATA #IMPLIED>                            ignore CDATA #IMPLIED>

<!ELEMENT include EMPTY>                        <!ELEMENT subblock EMPTY>
<!ATTLIST include                               <!ATTLIST subblock
  path CDATA #REQUIRED>                           name CDATA #REQUIRED
                                                  type CDATA #REQUIRED
<!ELEMENT blackbox EMPTY>                         desc CDATA #IMPLIED
<!ATTLIST blackbox                                ignore CDATA #IMPLIED
  addrbits CDATA #REQUIRED                        reps CDATA #IMPLIED
  desc CDATA #IMPLIED                             used CDATA #IMPLIED>
  name CDATA #REQUIRED
  reps CDATA #IMPLIED                           <!ELEMENT field EMPTY>
  type CDATA #REQUIRED                          <!ATTLIST field
  used CDATA #IMPLIED                             default CDATA #IMPLIED
  xmlpath CDATA #IMPLIED>                         desc CDATA #IMPLIED
                                                  ignore CDATA #IMPLIED
<!ELEMENT creg (field)*>                          name CDATA #REQUIRED
<!ATTLIST creg                                    trigger CDATA #IMPLIED
  name CDATA #REQUIRED                            type CDATA #IMPLIED
  default CDATA #IMPLIED                          width CDATA #REQUIRED>	


#### 2.2.1. **sysdef** Element and Its Children

The main element in the AGWB XML file is **sysdef**. It may contain an arbitrary number of child elements of three types: **include**, **constant**, and **block**.

The **include** element enables splitting the description into multiple files. It also allows creating the libraries of reusable XML definitions containing constants and blocks. When the system is built, the **include** element is replaced by the contents of the file pointed by its **path** attribute (This is a functionality added in the new version of AGWB.). An example of the system description is shown in Listing 2. The definition of a nested block that may be included in the system definition is shown in Listing 3.
sensors-21-07378-t001_Table 3Listing 2Example definition of the C&D-system. The example exposes various features of AGWB, which are explained in the next sections. Please note how constants are defined and used in expressions and attributes. The definition of **LINK_NR** constant exposes the limitations of XML—the expression “1<<LINK_NR_BITS” had to be coded as “1 &lt;&lt; LINK_NR_BITS”. The number of repetitions of block "I2C" is defined as a colon-separated list. That is associated with *variants* functionality described in Section 3.3.
<sysdef top="*MAIN*" masters="*2*">

<constant name="*NEXTERNS*" val="*4*" />
<constant name="*LINK_NR_BITS*" val="*5*" />
<constant name="*LINK_NR*" val="*(1 &lt;&lt; LINK_NR_BITS)-1*" />

<include path="*block1.xml*" />

<block name="*MAIN*" reserved="*1024*" >
  <blackbox name="*I2C*" type="*I2C_CTRL*" addrbits="*3*" reps="*8;4*" />
  <subblock name="*LINKS*" type="*SYS1*" reps="*LINK_NR + 1*" />
  <blackbox name="*BRAM*" type="*WB_BRAM*" addrbits="*12*" desc="*Block RAM*" />
  <creg name="*CTRL*" desc="*Control register in the main block*" >
    <field name="*LINK_SELECT*" width="*LINK_NR_BITS*" default="*7*" />
    <field name="*COUNT_MODE*" width="*4*" default="*2*" />
    <field name="*COUNT_RESET*" width="*1*" trigger="*1*" />
    <field name="*PLL_RESET*" width="*1*" trigger="*1*" />
  </creg>
  <creg name="*TEST_OUT*" width="*17*" reps="*3*" default="*23*" stb="*1*" />
  <sreg name="*TEST_IN*" width="*16*" reps="*4*" ack="*1*" />
</block>

</sysdef>

sensors-21-07378-t001_Table 4Listing 3Example definition of the nested block. Please note that this is not a valid AGWB XML description. It must be included in the system definition to form the valid AGWB XML description.
<block name="*SYS1*" aggr_outs="*1*" >
  <creg name="*CTRL*" desc="*Control register*" stb="*1*" >
    <field name="*START*" width="*1*" trigger="*1*" desc="*Start the operation*" />
    <field name="SPEED" width="*4*" default="*-1*" type="*signed*"
           desc="*Transmission speed*" />
    <field name="*STOP*" width="*1*" trigger="*1*" desc="*Stop the operation*" />
  </creg>
  <sreg name="*STATUS*" desc="*Status register*" ack="*1*" >
    <field name="*RX_AV*" width="*1*" desc="*Received data available*" />
    <field name="*TX_RDY*" width="*1*" desc="*May accept data to transmit*" />
    <field name="*TX_DONE*" width="*1*" desc="*All data transmitted*" />
    <field name="*TX_ERROR*" width="*2*" desc="*TX error code*" />
    <field name="*RX_ERROR*" width="*4*" desc="*RX error code*" />
  </sreg>
  <sreg name="*RXD*" desc="*Received data register*" ack="*1*" />
  <creg name="*TXD*" desc="*Transmit data registers*" stb="*1*" default="*0x0*" />
</block>


The **constant** (This is a functionality added in the new version of AGWB.) element defines a constant with value given by its **val** attribute. That value may be a result of simple Python expression, as shown in Listing 2. The constants are passed to the generated VHDL and software support codes.

The **block** describes a block of the system, as described in Section 1.2.

The **sysdef** element must have a **top** attribute that should be set to the name of the block being the top of the design.

In many applications, the C&D-system may have a few bus masters. For example, the Wishbone bus may be controlled via an IPbus interface or via a JTAG-based controller, which is slower but enables debugging when the network connection is not available yet. Another master may be a soft CPU included in the design, which should perform the system’s initial configuration, making other interfaces operable. If the system has more than one master, the **sysdef** should have an optional **masters** attribute set to the number of masters.

#### 2.2.2. **block** Element

The **block** element defines a block of the C&D-system. It may contain subblocks (defined by **subblock** child elements) and registers. The control registers are defined by **creg**, and status registers by **sreg** child elements. As stated in Section 1.2.3, the subblocks and registers may be instantiated as vectors. In that case, their **reps** attribute should be set to the number of elements in the vector.

The block may also contain a subblock that is not generated by AGWB. In that case, it is instantiated as a special **blackbox** child node with obligatory **addrbits** attribute defining its number of address bits.

Both **subblock** and **blackbox** elements must have a **type** attribute defining their type. In the case of **subblock** it is the name of the instantiated **block**. In the case of the **blackbox**, it is just the user-defined name of the type.

The advanced functionalities in the block description are the possibility to reserve certain space at the beginning of the block address space with the **reserved** attribute, and aggregating the registers’ input and output signals into VHDL records with **aggr_ins** and **aggr_outs** attributes. Setting the **testdev_ena** attribute to non-zero inserts the *test device* into the generated block. It contains 4 registers enabling verification of correct bus operation and error detection.

#### 2.2.3. Description of Registers

The control (read/write) and status (read-only) registers are described with very similar **creg** and **sreg** elements. The user may define their **width** (1 to 32 bits) and **type** (std_logic_vector, signed or unsigned).

The register may be split into bit-fields. In that case, it contains **field** child elements (see Section 2.2.4).

The status registers may be connected to the additional *acknowledge* signal asserted for one clock pulse, whenever the register is read. That requires setting the optional **ack** attribute to non-zero value. Similarly, the control registers may be connected to additional *strobe* signal by setting the optional **stb** attribute to a non-zero value. The signal will be asserted for one bus clock period whenever the register is written (This functionality has been improved in the new AGWB.). Those signals may be used as read and data strobes when the register is connected to a FIFO block.

For control registers, their initial value may be set with the **default** attribute (The **stype** attribute is used to avoid VHDL namespace collisions in specific situations outside the scope of this paper.).

#### 2.2.4. Description of Bit-Fields

The **field** elements define bit-fields inside of registers. The **width** attribute defines the number of bits in the bitfield. The sum of widths of all fields in the register must not exceed 32. The **field** element in a control register may also have the **default** attribute. The bitfield in the control register may have a **trigger** attribute (This is a functionality added in the new version of AGWB.). With this attribute set to the non-zero value, all bits in that bitfield, when written with ‘1’, remain asserted only for a single period of the bus clock. They are always read as zeroes. Such bit-fields are dedicated to triggering actions in the hardware.

### 2.3. Address Allocation Algorithms

Work with the **addr_gen** (see Section 1.4.3) has shown that assigning subsequent addresses to the registers and blocks without proper alignment resulted in suboptimal address decoders. That increased resource consumption and the critical path length, resulting in lowering of maximum bus clock frequency.

Therefore, AGWB uses another address allocation algorithm oriented on optimization of address decoders.

If the block needs *M* bits for internal addressing, it is aligned in the address space to the $2^{M}$ boundary. It ensures that the access to the block may be easily decoded by simple binary comparison of address bits *M* to 31. That task is performed by the Wishbone crossbar to which the block is connected.

The address allocation algorithm also minimizes the occupied address space by avoiding unnecessary fragmentation:For each block, the number of used addresses *L* is calculated as the sum of sizes of its registers and subblocks. The number of addresses is rounded up to the nearest power of two: $P=2^{N}$, where *N* is the smallest integer for which $2^{N}\geq L$. The *P* value is the size of the block. That step may be performed recurrently, as the sizes of subblocks are needed to calculate the size of the parent block.In each block, the registers are located at the beginning of the address space (If there is a reserved area at the beginning, defined with the **reserved** attribute (see Section 2.2.2), it will be placed before the registers.) (so **ID** and **VER** have well-defined locations). The subblocks are sorted according to their decreasing size and are placed starting from the end of the block’s address space.The final address map is built starting from the top block located at address 0 and traversing its subblocks.

Such an algorithm ensures that each block’s base address is properly aligned and warrants minimal address space fragmentation.

## 3. Usage of the AGWB C&D System Generator

Processing of the AGWB C&D-system description and generating of the output files is performed by executing the **addr_gen_wb.py** script. Usage in the design flow and possible command-line options are shown in Figure 5.

The AGWB for each block generates the following VHDL files:the **BLOCKNAME.vhd** providing the AGWB local node (ALN) code,the **BLOCKNAME_pkg.vhd** providing the **BLOCKNAME_pkg** package containing the types and constants specific to that block,for the top block only, the **BLOCKNAME_const_pkg.vhd** containing the constants defined in the system definition XML file.

The block-specific constants belong to the respective packages. That enables avoiding name conflicts, which may, for example, occur when registers with the same name (and hence of the type with the same name) are used in multiple blocks.

To facilitate integration of the generated files with the HDL project, the AGWB may generate files for one of two build systems: FuseSoc [47] or VEXTPROJ [48]. Further synthesis and implementation of the FPGA firmware is performed in the appropriate vendor-specific environment (e.g., Vivado for Xilinx FPGAs).

### 3.1. Integration of AGWB-Generated Part with User Logic

The C&D-system components generated by the AGWB must be properly embedded in the user logic. The user must implement the appropriate connection of the master Wishbone buses to the masters and route the child node Wishbone buses to the nested blocks. The simplified diagram of the necessary connection is shown in Figure 6.

The AGWB has been designed in a way that simplifies the integration. For example, for it may create special types or aliases for individual registers or vectors of registers, to minimize the risk of unnoticed mistakes.

Connections between the blocks are intended to be done with standard Wishbone buses. The appropriate types defining a single bus or a vector of buses are provided by the library General cores [22] and used by AGWB.

#### Use of C&D System in Designs Using Multiple Clock Domains

Different data processing blocks may work with different clock frequencies in more complex data acquisition and processing systems. If the Wishbone bus working with the same clock is used across the whole design, reading and writing the registers driven by another clock requires proper synchronization. Doing it at the register level would be very resource-consuming. A better approach may be to create the Wishbone bus segments operating at different clock frequencies and connect them via an appropriate clock-domain-crossing (CDC) block. AGWB offers a dedicated CDC block optimized for use in the C&D-systems.

However, if the blocks working with different clocks are scattered throughout the design, there are two approaches for splitting the control bus.

The first approach preserves the logical organization of the control tree and uses CDC whenever the part of the block is operating with another clock. That approach is shown in Figure 7.

The advantage is that each block may use a coherent area in the address space. The cost is that the number of CDC blocks is high. This approach may lead to connecting a few CDC in series in certain topologies, which results in a significant slow-down of register accesses (each synchronization stage consumes a certain number of clock cycles during the transaction).

Another approach is shown in Figure 8.

In this approach, the buses working with different clocks are independently routed through the design. That keeps the minimal possible number of CDC blocks but increases resource consumption for routing the two independent buses. Additionally, the block may occupy two or more separate areas in the control address space, complicating the software’s writing.

The selection of the right approach is the responsibility of the user. Sometimes the best solution could be using the first approach in certain parts of the design, and the second in others.

### 3.2. Integration of C&D System with Software

The hardware part of the C&D-system connected to the user’s logic must be integrated with the software. That requires a few main points:The software must be informed about the structure of the hierarchy, addresses and properties of the individual registers (obligatory);There may be a hierarchy of objects created that reflects the hierarchy of blocks and registers in the software (optional)

The method to fulfill those requirements may depend on the language in which the software should be written.

Currently, there are a few languages supported, which are handled by the appropriate backend routines generating the necessary files.

#### 3.2.1. IPbus Backend

The AGWB development was started for the systems controlled via the IPbus interface [23]. Therefore, the generation of the address map in the XML format suitable for IPbus [39] was an initial solution. The address tables generated by the IPbus backend may be used by Python software and by C++ software. However, the IPbus address table format has a significant disadvantage—it does not support vectors of registers nor vectors of blocks. Each element of the vector must be specified individually in the address table.

The advantage of the IPbus address table is that its modification does not require recompilation of the software. For example, the same compiled software may support hardware with different versions of the firmware. It is sufficient to load the appropriate version of the address table.

#### 3.2.2. AMAP XML Backend

The AMAP XML format has been created to avoid the limitations of the IPbus format while preserving its advantages. It has been extended with support for vectors of registers and vectors of blocks. Therefore, the nodes may have two additional attributes:**nelems**—describing the number of elements in the vector,**elemoffs**—the distance in the address space between the base addresses of the consecutive elements of the vector.

Additionally, it introduces different XML elements for different types of nodes (the IPbus XML address table keeps everything in an element **node**). The block definition is stored in the element **module**. The subblock instances are stored in the elements **block**. The register definitions are stored in the elements **register**.

The software using the AMAP XML, similarly to the IPbus software, should allow loading or reloading the definition of the system. Therefore, the modification of the address map does not enforce recompilation of the software.

The AMAP XML format supports **variants** (described in Section 3.3).

#### 3.2.3. Native Python Backend

The native Python backend gives the best integration with the Python language. The hierarchy of blocks and registers is fully reflected in the hierarchy of classes. The structure of the C&D-system is reflected in the tree of objects created when the user imports the generated Python code and accesses the register.

The native Python mode is very flexible. It may use different connections to the Wishbone bus. For elementary access, the user must only define a virtual interface with read(self, address) and write(self, address, value) methods.

AGWB may be used with network-based control interfaces, which offer high throughput, but high round-trip latency. It may significantly affect the performance of the read-modify-write (RMW) operations, often used in control algorithms. Performing the RMW operation in hardware and aggregating multiple bit-field operations into a single RMW operation may cure that problem (A good example may be an Ethernet-based IPbus [23] interface which implements both mentioned optimizations.). The native Python backend may use those optimizations if a few additional methods are implemented: writeb(self, address, value) and readb(self, address) for scheduling the read and write operations, write_masked(self, address, mask, value), and writeb_masked(self, address, mask, value, more=False) for read-modify-write operations (immediate or scheduled), and dispatch() that executes the scheduled operations.

The native Python mode is ideal for interactive debugging that involves the software running on a PC. The user may directly access the registers. The Python introspection and reflection functionalities are available in the interactive mode. The complex Python routines also may be executed.

#### 3.2.4. C Header Backend

For software written in C, AGWB generates a few C headers. The constants defined for the whole system are placed into the dedicated **agwb_TOPBLOCK_const.h** file. The constants are generated as numerical values, but they are accompanied by a comment explaining how that value was calculated in Python (e.g., #define NSEL_MAX 31 // (1 << NSEL_BITS)-1).

For each block, there is a related header **agwb_BLOCKNAME.h** generated. This header contains the block’s ID and VER values (see Section 2.1), definition of the structure defining the layout of registers, and inline functions for reading or writing the values of the bit-fields.

The C backend is generally dedicated to writing the C drivers (especially the kernel drivers) for systems where the FPGA-implemented register is directly mapped into the CPU address space. Typical use cases are the SoC or MPSoC with FPGA part connected directly to the AXI bus of the CPU (in that case, an AXI to Wishbone bridge is needed) or system with FPGA connected via PCIe interface (in that case also an appropriate bridge PCIe-AXI-WB is needed).

The advantage of the C backend is fast and direct access to the C&D-system registers. Its disadvantage is a necessity to recompile the software whenever the design is modified.

#### 3.2.5. Forth Backend

The Forth backend enables control of the AGWB-generated system from the J1B Forth CPU [49,50,51]. It is a simple synthesizable CPU able to execute the programs written in Forth [52] language.

The Forth language supports very efficient interactive work, so it is a good tool for interactive debugging. It requires only a console connection (Usually, Forth debugging uses the UART, but it is also possible to provide a console via JTAG, SPI, or another interface.) to the FPGA with firmware containing J1B. During the interactive work, the programmer still may define procedures (called “words” in Forth) and create fairly complex algorithms. It is also possible to put the defined procedures into the FPGA configuration bitstream, enabling automated execution of a certain word after the system starts. Therefore, J1B may be a perfect tool for the initialization of the board.

The J1B CPU is optimized for low resource consumption and, therefore, it has limited code memory. In the case of a complex AGWB-generated C&D-system, its memory may become filled just with the names of blocks and registers. For initialization, usually, only a small subset of the registers is needed. The unused blocks and registers may have the optional **ignore** attribute set to forth to be excluded from the generated Forth software.

### 3.3. Support for Special Types of Designs

The AGWB assumes that the instances of the block are identical. It was a conscious design decision that simplifies the allocation of addresses and generation of code. That is why no user parameters are passed to the **subblock** element. Therefore, the size allocated for the block in the address space is always the same.

However, there may be situations where not all resources provided by certain instances of the block are needed.

An example may be the usage of multigigabit transceivers (MGT). They are often grouped in banks containing four transceivers, which share certain infrastructure. Therefore, it makes sense to create an MGT controller block in C&D-system that controls four MGTs. Let us assume that our FPGA has four such banks, but we need to use two MGTs for other purposes and require another controller. So, we need three instances of MGT controller that control four MGTs, and one instance which controls only two MGTs.

AGWB handles such cases at the level of integration of the AGWB-generates C&D-system with the user logic. In theory, we could only not connect two MGTs to the last instance of the MGT controller. Unfortunately, that approach does not warrant that unused resources will be optimized out during the firmware synthesis (The control registers are storage elements, so they remain synthesized even if they are not connected to any user logic. Similarly, the CDC blocks must be protected against certain optimizations, resulting in keeping them even if they are not used.).

#### 3.3.1. Precise Customization

To avoid it, AGWB offers a **precise customization** functionality (This is a new functionality added in the new version of AGWB.), enabling the developer to control which registers or subblocks and how many of them are synthesized. The generated ALN code uses special VHDL generics of form **g_OBJECTNAME_size**. Those generics may be set to 0 (excluding the related object from synthesis) or 1 (including it) for single blocks or registers. For vectors of blocks and registers, those generics may be set to 0 (excluding the whole vector from synthesis) or to any value between 1 and the maximum size (defined with the **reps** attribute described in Section 2.2.2). That maximum size is defined as the constant **c_OBJECTNAME_size** in the generated VHDL package and is used as default values of the related generic. The user not interested in using precise customization **precise customization** may omit the generic during instantiation, and it will remain set to its maximum value.

Of course, the above modifications affect the hardware part of the generated C&D-system. The user’s responsibility is to ensure that the C&D software is aware of the irregularities handled by precise customization.

#### 3.3.2. Design **Variants**

Usage of AGWB with big Xilinx FPGAs based on Stacked Silicon Interconnect (SSI) technology [53] and divided into multiple Super Logic Regions (SLRs) [54] have exposed yet another need for handling irregularities in the AGWB-described C&D-system.

The communication between SLRs requires special Super Long Lines (SLL) [54], a scarce resource. Therefore, if the FPGA has two SLRs (That is the case of the xcku115-flvf1924 FPGA available in the CRI boards used in the CBM experiments.), where each is connected to the host via a separate PCIe interface, the optimal solution may be to implement two similar data acquisition and processing subsystems—one in each SLR.

Certain blocks, however, must be implemented only in one SLR and communicate with another SLR via SLLs. That introduces irregularities of another kind than those described in Section 3.3.1. There, the irregularities appeared in the same C&D-system. Here, we need to put two different versions of the C&D-system into the same FPGA.

Due to the limitations of VHDL, there is no simple solution of such problem. The finally implemented and working solution extends the precise customization described in Section 3.3.1 and is called **variants** (This is a new functionality introduced in the new version of AGWB.).

The user may define a few **variants** of the design, and specify the values of certain attributes for each variant individually. That is achieved by defining a colon-separated list of possible values instead of a single value, like below:


<blackbox name="I2C" type="I2C_CTRL" addrbits="3" reps="8;6;4" />


The above description defines three variants of the design, where variant 0 has 8 I2C controllers, variant 1–6, and variant 2–4 controllers.

Of course, all variant-dependent attributes must define the same number of variants. The following description will result in an error:


<blackbox name="SPI" type="SPI_CTRL" addrbits="3" used="1;0" />


because the first line defines three variants, while the second only two.

When using variants, the additional constants **v_OBJECTNAME_size** are defined in the generated package, which are the integer arrays, storing the size of the object in each variant.

Currently, only two software backends support **variants**:native Python (see Section 3.2.3),AMAP XML (see Section 3.2.2).

If the block is variant-dependent and the C&D-system is supposed to be used with the variants-aware software, the user should set the VER value for the variant-dependent value (calculated as a CRC32 of the generated AMAP XML description of that block). The following setting of generic provides that during the instantiation of the block:

g_ver_id => v_BLOCKNAME_ver_id(variant_number).

The number of the variant should be propagated throughout the design as a dedicated integer generic because a variant-dependent block may be used on a lower level of hierarchy.

The example of the instantiation of the variant-dependent block is shown in Listing 4.
sensors-21-07378-t001_Table 5Listing 4Example of instantiation of the variant-dependent ALN.
 [...]

 -- *LINKS are a vector of blocks so they use a vector of Wishbone buses*
 signal LINKS_wb_m_o : t_wishbone_master_out_array(v_LINKS_size(var_nr)-1 downto 0);
 signal LINKS_wb_m_i : t_wishbone_master_in_array(v_LINKS_size(var_nr)-1 downto 0);
 -- *EXTHUGE is a single block present in only one variant*
 signal EXTHUGE_wb_m_o : t_wishbone_master_out;
 signal EXTHUGE_wb_m_i : t_wishbone_master_in;
 -- *TEST_IN is a variant-dependent vector*
 signal TEST_IN_i : ut_TEST_IN_array(v_TEST_IN_size(var_nr)-1 downto 0)
        := (others => (others => '*0*' ));
 -- *The below signals are related to the variant-independent blocks and registers*
 signal EXTERN_wb_m_o : t_wishbone_master_out_array((C_NEXTERNS-1) downto 0);
 signal EXTERN_wb_m_i : t_wishbone_master_in_array((C_NEXTERNS-1) downto 0);
 signal CDC_wb_m_o : t_wishbone_master_out_array((C_NEXTERNS-1) downto 0);
 signal CDC_wb_m_i : t_wishbone_master_in_array((C_NEXTERNS-1) downto 0);
 signal CTRL_o : t_CTRL;
 signal TEST_OUT_o : t_TEST_OUT_array := (others => (others => '*0*' ));

 [...]

 MAIN_1 : entity agwb.MAIN
   generic map(
     g_ver_id => v_MAIN_ver_id(var_nr),
     g_LINKS_size => v_LINKS_size(var_nr),
     g_EXTHUGE_size => v_EXTHUGE_size(var_nr),
     g_registered => 2
)
   port map (
     slave_i        => wb_s_in,
     slave_o        => wb_s_out,
     LINKS_wb_m_o   => LINKS_wb_m_o,
     LINKS_wb_m_i   => LINKS_wb_m_i,
     EXTHUGE_wb_m_o => EXTHUGE_wb_m_o,
     EXTHUGE_wb_m_i => EXTHUGE_wb_m_i,
     EXTERN_wb_m_o  => CDC_wb_m_o,
     EXTERN_wb_m_i  => CDC_wb_m_i,
     CTRL_o         => CTRL_o,
     TEST_IN_i      => TEST_IN_i,
     TEST_OUT_o     => TEST_OUT_o,
     rst_n_i        => rst_n_i,
     clk_sys_i      => clk_sys_i);

 [...]



## 4. Results-Practical Applications

Usage of AGWB requires connection to a compatible hardware interface, supporting control of the Wishbone bus from the software. Up to now, the following hardware interfaces have been prepared and tested.

The simplified **cbus.py** interface supporting only read and write commands. It enables testing of AGWB-generated C&D-system in simulation with GHDL.Connection to the IPbus master, with minimal signals adjustments (as described in [55]). The IPbus may be used with AGWB in two ways:-using the standard C++ and Python libraries designed for IPbus (with the code generated by the IPbus backend in Section 3.2.1),-using only the “client” object offered by the IPbus library and its access procedure. This method works with the native Python backend (see Section 3.2.3).The interface controlling Wishbone bus via JTAG interface [56].The PCIe interface based on Xilinx “AXI Bridge for PCI ExpressGen3 Subsystem v3.0” [57] and simple AXI-Lite to Wishbone bridge. The solution is available in the project [58] (branch “agwb”). The solution based on the AXI-Lite to Wishbone bridge may also be directly used to connect the AGWB-based system to an ARM CPU available in the SoC or MPSoC systems.The dedicated PCIe to Wishbone bridge developed for the CBM experiment [59]. That interface allows operation in “native Python” mode and in “AMAP XML” mode.A specialized GBT-SC [60] interface which, when used together with the GBT-FPGA [4] core, enables control of the Wishbone bus via the GBT [61] link.

Up to now, the AGWB has been successfully used in a few experimental projects [50,58,62] and four serious projects used for CBM and BM@N experiments:the DPB firmware [37],the GBTX Emulator (GBTxEMU) [63,64],the SMX tester [59],the CRI firmware [59].

Intensive use of AGWB in real projects and feedback received from other developers and users contributed to introducing new functionalities and eliminating bugs. The DPB firmware [37] was the first practical project using AGWB. In the GBTX Emulator (GBTxEMU) [63], AGWB is used with three different masters—the IPbus, the J1B, and a project-specific GBT IC master. This project enabled testing and has proven the correctness of the multi-master functionality. It also required the use of the CDC block. The CRI firmware [59] project was the most demanding for AGWB up to now. The software uses extensive hardware detection and testing functions, which require reliable support for bus errors and timeouts. Its development and usage resulted in significant improvement and intensive testing of the CDC block and the introduction of the built-in test device. The CRI firmware is used in the big FPGAs consisting of two SLRs resulting in introducing the “variants” functionality described in Section 3.3. The designs prepared for the CBM experiment share significant parts of the C&D-system. Therefore, their development was a good test of the reusability of fragments of the AGWB system description.

## 5. Discussion and Conclusions

The AGWB fulfills all the requirements set up for the C&D-system generator in Section 1.2 and listed in Table 1. Contrary to SystemRDL, the AGWB from the very beginning was created simultaneously as the description format and as a conversion tool. Therefore, all proposed functionalities were immediately analyzed concerning the viability of implementation. A good example was adding the support for precise customization and variants. This approach resulted in a good balance between the functionality available to the user, low complexity of implementation, and simplicity of description.

The AGWB appeared to be a useful tool for generating C&D-systems. The syntax of the C&D-system description is simple and may be easily edited in any text editor. The nesting depth in the description is limited to three (block, register, field), regardless of the number of hierarchy levels in the design. Therefore, even complex hierarchical systems may be described with legible XML files.

Good support for parameterization of the design has been confirmed in practical use, such as changing the number of components for simplified debugging versions. It has been confirmed that properly used AGWB indeed minimizes the effort needed to adjust the user logic to the modified parameters.

The possibility to include the fragments of XML files facilitates sharing of the definitions between different C&D-systems.

Adding new registers and blocks is relatively simple. The workflow is friendly for text-based tools, end hence for version control systems.

Of course, AGWB has its limitations resulting from the compromise between functionality and simplicity. It does not offer the versatility and completeness of SystemRDL, but is a small and consistent solution.

It is a fully open-source solution. The source code is available in the GitHub repository [65]. The LGPL V2 license allows the user to modify it for his or her needs, and the system’s simplicity should facilitate such modifications.

The code generated by the AGWB may be freely used and distributed by the user, but it relies on the General cores library and uses the components licensed under Solderpad Hardware License, Version 2.0 [66].

### Future Plans

The AGWB system has been developed as a tool supporting currently developed systems. Its gradual evolution resulted in a structure where analysis of the system description is somehow interconnected with the output generation. In future versions, it may be advantageous to improve that separation by formalizing the internal representation of the generated system. That may motivate users to create their own software or hardware backends.

The end users suggested improvements to add support for local constants and pass user arguments to block instances (subblocks). Such extensions may increase the usability of AGWB, but they must be carefully tested regarding the possibility of implementation. For example, making the block’s address space size dependent on the user-provided parameters would significantly complicate the address allocation algorithm and the generated ALN code.

It is expected that the role of SystemRDL will increase in the future. Therefore, it should be investigated if it is possible to support conversion between the AGWB description and a certain well-defined subset of SystemRDL.

## Figures and Tables

**Figure 1 sensors-21-07378-f001:**
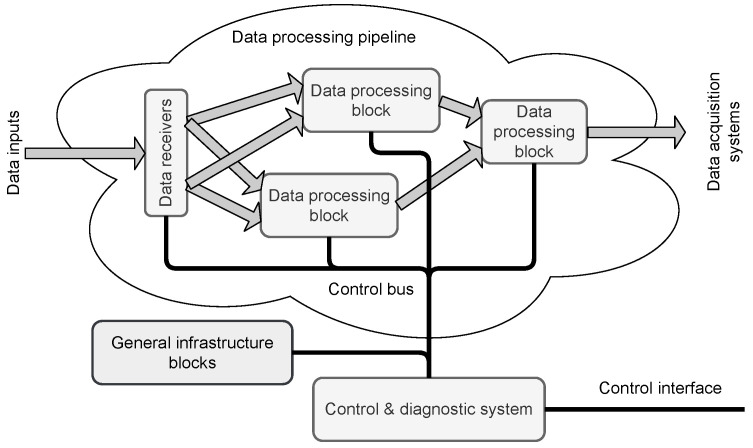
The general block diagram of the data acquisition and preprocessing system implemented in FPGA. The acquired data are concentrated and preprocessed in the pipelined datapath before sending to the central data acquisition (DAQ) system. The Control and Diagnostics (C&D) system uses an independent bus to access the registers located in different parts of the data processing pipeline and general infrastructure blocks.

**Figure 2 sensors-21-07378-f002:**
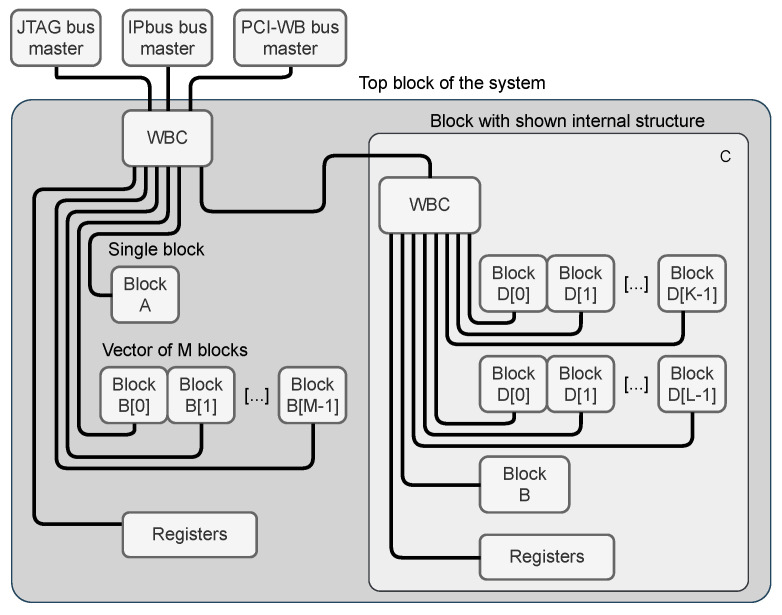
The simplified block diagram of the AGWB-based C&D-system in the FPGA. The “WBC” blocks are Wishbone crossbars.

**Figure 3 sensors-21-07378-f003:**
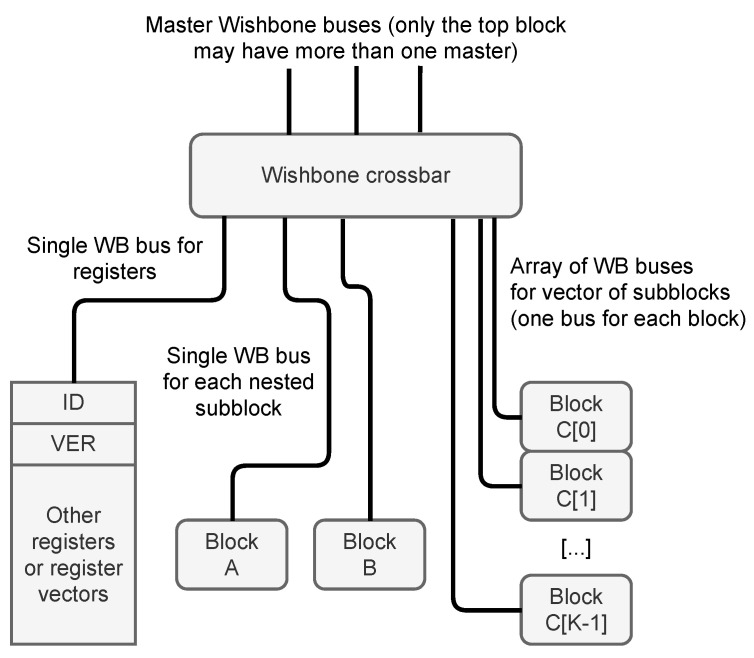
Allocation of Wishbone buses for registers and child nodes (subblocks).

**Figure 4 sensors-21-07378-f004:**
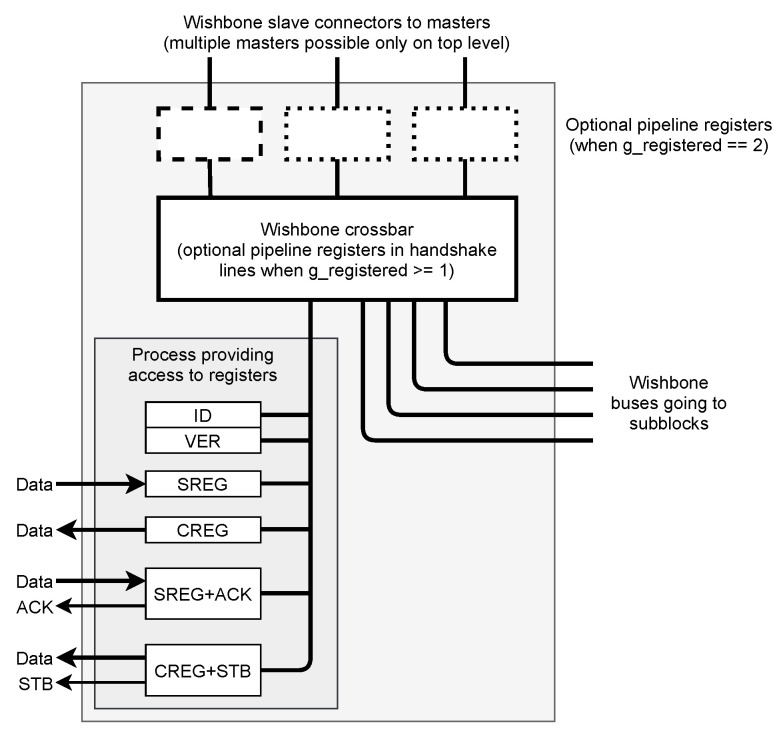
The structure of the “AGWB local node” (ALN) automatically generated by AGWB. ALN should be placed by the user in the design, as described in Section 3.1.

**Figure 5 sensors-21-07378-f005:**
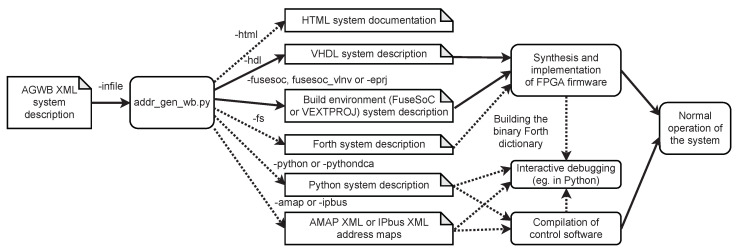
Usage of the **addr_gen_wb.py** script. Available command-line options are listed at arrows. Dotted arrows denote the optional output.

**Figure 6 sensors-21-07378-f006:**
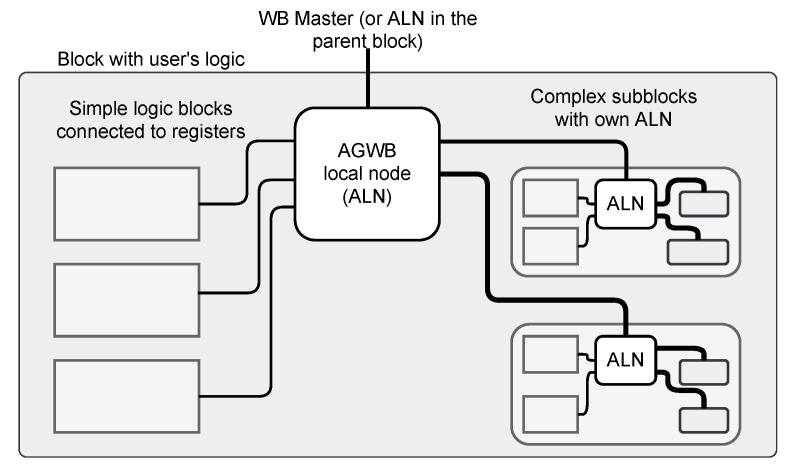
Block diagram of the single block, showing how the signals generated in the AGWB local node (ALN) should be connected to the user’s logic consisting of simple logic blocks connected to registers and complex subblocks with their own ALN. The structure of the ALN is shown in Figure 4.

**Figure 7 sensors-21-07378-f007:**
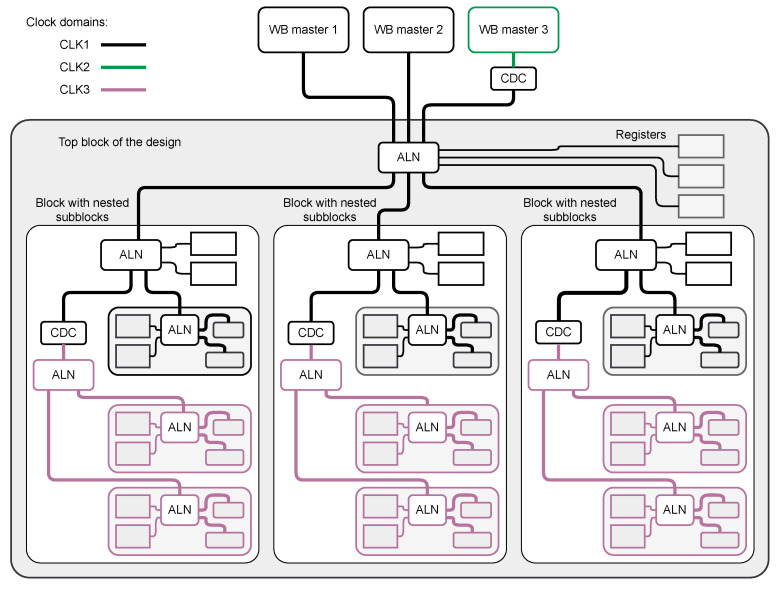
Block diagram of a system with multiple clock domains. One of the WB masters works with another clock domain. Each of the top block subblocks has a part that works with another clock. In this approach, each block is controlled via a single Wishbone bus. For the part working with another clock, the internal CDC is implemented.

**Figure 8 sensors-21-07378-f008:**
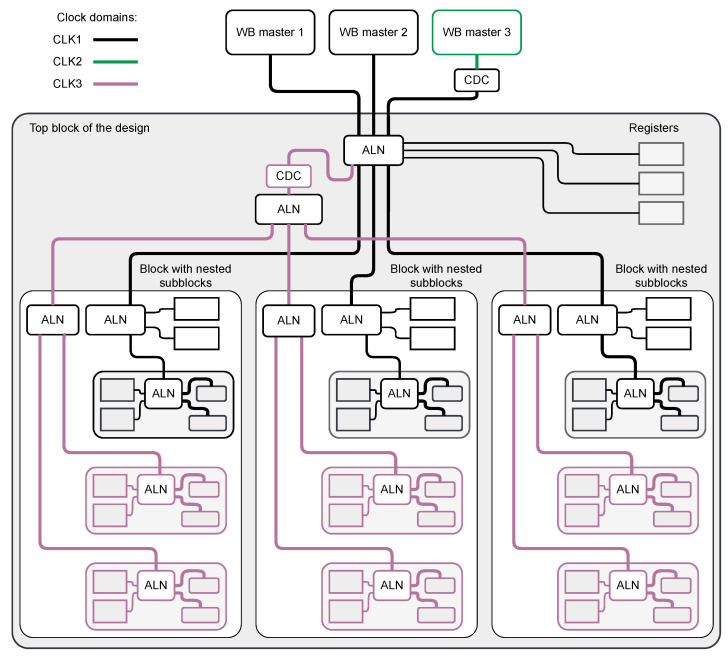
Block diagram of a system with multiple clock domains. One of the WB masters works with another clock domain. Each of the top block subblocks has a part that works with another clock. In this approach, for two clocks used in the Wishbone slaves, the separate buses are created at the top level. This split of control bus needs only one additional CDC, but two buses must be routed through the design, and the address space of each subblock is split into two areas.

**Table 1 sensors-21-07378-t001:** Results of comparison of available tools supporting C&D-system generation. None of the available tools fulfills all requirements formulated in Section 1.2.

	Support for Multilayer Hierarchical Systems	Support for Parameterized Designs	Support for VHDL Language	Support for Wishbone Bus	Support for Bit-Fields	Generation of the Register Access Code	Free and Open Source
SystemRDL	yes	yes	no open tools yet	no open tools yet	yes	no open tools yet	yes
II & CII	yes	yes	yes	yes (in CII)	yes	yes	no
addr_gen	yes	yes	yes	supports Ipbus—very similar to WB	no	no	yes
wbgen2	no	no	yes	yes	yes	yes	yes
Opentitan register tool	no	no	no	no	yes	yes	yes
hdlregs	no	no	yes	yes	yes	yes	yes
RgGen	no	no	yes	no	yes	yes	yes
rgen	no	no	no	no	yes	unclear	yes
cheby	yes	no	yes	yes	yes	yes	yes

## Data Availability

Not applicable.

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
