# Peer review of "Control and Diagnostics System Generator for Complex FPGA-Based Measurement Systems"

_sensors, 2021, doi:10.3390/s21217378_

Round 1

Reviewer 1 Report

In “Control and Diagnostics System Generator for Complex FPGA-based Measurement Systems”, authors present “a new solution based on the XML system description, facilitating the automated generation of the control system’s HDL code and software components and enabling easy integration with the control software.”

Main advantage of such an approach, they say, are: reusability, ease of maintenance, easy detection of mistakes, and the possibility of use in modern FPGAs.

The methodology they developed is already been tested in a DAQ system used in a HEP experiment – the system code is available on GitHub as open source solution.

As they say, at the beginning of sec.2, the generation of such a FPGA’s systems is not a new problem and many solutions have been developed.

In effect, the same authors have already proposed a solution for this problem [46], producing an automatic system for the management of FPGA local bus but, in the current work, “the system has been significantly improved”.

The core of the paper is the development of a AGWB XML system description, that enable an automated generation of the control system’s HDL-code and simplify the interface with software components.

This work is developed in a very specific framework (HEP experiment) and the authors have a long and consolidated experience in this field – they manage with great confidence a number of very specific design tools and methodologies.

The work is well described and rich of suitable references.
In my opinion, although definitely too long, the work is practically ready for the publication.

Nevertheless, I have few observations I hope could stimulate some integrations by authors:

-    I know well that in this case is not simple to present quantitative performance that can be useful to compare this approach with others one but, at least, the differences between this version and the previous work [46] should be highlighted better – for example, using a table to describe the main function and compare the results of the last solution with the previous one, and with other present in literature; this can semplify the valutaion of the reader;

-    In the paper, no specific model of FPGA is cited – authors speak generically of complex Xilinx FPGA - It would seems that such an approach was developed using the MicroBlaze CPU (see note 2 at p.2) – what about this work using a modern SoC FPGA in which Complex FPGA are integrated with RISC 32bit ARM Processor? is this approach useful even using these technologies? In my opinion, providing more information and details on the actual implementation is desirable.

-    Finally, in my opinion, the process of compilation/synthesis of circuit/interface starting from AGWB XML is not described with sufficient clarity – a picture with the design-flow can be clarify better the whole procedure.

Reviewer 2 Report

The paper describes a novel tool for data acquisition of FPGA-based measurement systems.  The proposed solution connects to an open-source realization of the tool, which can be downloaded from Github, so the proposed solution seems a very nice application which works and freely available.

However, I have major issues with the paper. Mainly the structure of the text. The introduction of the papers, before the bullet points is very short and doesn't tell any information to the reader that what is the difference between the current solution and the other existing solutions in the field. Some concrete examples should be listed and the novelty of the proposed solution should be stated at the end of the introduction.

I think this first section should be restructured, shortened to give a brief overview and make the context for the reader. This text goes too fastly into deep details, this is my other point is that some focus points, should be highlighted and those points discussed in the other parts, because this part contains too much commenting on an example code, so the text looks like a catalogue or some marketing material, which is not a problem, but some comparison with other methods or validation is necessary in scientific writing.

Round 2

Reviewer 2 Report

The authors answered all of my questions.